# Phenolic Compounds in Active Packaging and Edible Films/Coatings: Natural Bioactive Molecules and Novel Packaging Ingredients

**DOI:** 10.3390/molecules27217513

**Published:** 2022-11-03

**Authors:** Ajit Kumar Singh, Jae Young Kim, Youn Suk Lee

**Affiliations:** Department of Packaging, Yonsei University, Wonju 26393, Korea

**Keywords:** polyphenols, active packaging, edible film/coatings, bioactive compounds, antioxidants, antimicrobials, food safety

## Abstract

In recent years, changing lifestyles and food consumption patterns have driven demands for high-quality, ready-to-eat food products that are fresh, clean, minimally processed, and have extended shelf lives. This demand sparked research into the creation of novel tools and ingredients for modern packaging systems. The use of phenolic-compound-based active-packaging and edible films/coatings with antimicrobial and antioxidant activities is an innovative approach that has gained widespread attention worldwide. As phenolic compounds are natural bioactive molecules that are present in a wide range of foods, such as fruits, vegetables, herbs, oils, spices, tea, chocolate, and wine, as well as agricultural waste and industrial byproducts, their utilization in the development of packaging materials can lead to improvements in the oxidative status and antimicrobial properties of food products. This paper reviews recent trends in the use of phenolic compounds as potential ingredients in food packaging, particularly for the development of phenolic compounds-based active packaging and edible films. Moreover, the applications and modes-of-action of phenolic compounds as well as their advantages, limitations, and challenges are discussed to highlight their novelty and efficacy in enhancing the quality and shelf life of food products.

## 1. Introduction

Considering socioeconomic developments such as urbanization, globalization, changing professions and lifestyles, and rising incomes, it is evident that the next generation of consumers prefers packaged foods. As a result, it is anticipated that the food processing industry will experience major challenges in the near future in providing healthy and safe food [1]. Traditionally, food packaging has been considered an essential aspect for protecting fruits, vegetables, meats, dairy products, and processed foods as well as extending their shelf lives. Containment, protection, communication, and convenience are among the primary roles of food packaging [2,3]. However, these traditional functions of packaging are often passive and insufficient to maintain the quality of food products, particularly those requiring packaging materials with high gas permeability or perforations to permit vapor/gas transfer through the packaging, such as fresh agricultural and meat products. With shifting consumer preferences and demands, the research and development of packaging has dramatically expanded its scope beyond its primary functions. However, the use of chemical compounds as additives in food packaging materials is unsuitable because of their high toxicity and detrimental effects on both human health and the environment [2,4]. Additionally, concerns over food safety and the recognition of the associated health risks received considerable recent attention, which has led to an increase in the use of clean-label foods; these are minimally or never processed, do not contain chemical additives, and have standard shelf lives [4]. In this context, the current emphasis is on designing innovative packaging films and edible coatings integrated with active substances that can effectively inhibit oxidation and microbial spoilage [5].

Active packaging is a pioneering approach for developing packaging systems that provide dynamic protection to foods and achieve quality control primarily via advantageous food-packaging interactions. According to European Regulation (EC) No. 450/2009, active-packaging schemes are defined as “packaging systems that dynamically integrate constituents which would emit or absorb elements into or from the packaged foods or the ambient atmosphere of the food” [6]. Typically, active-packaging systems perform actions such as absorbing, emitting, extracting, controlling, and assisting foodstuff states as needed from inside or outside. Among the components that can be embedded in packaging materials, moisture absorbents, gas scavengers or emitters, gas flushers, ethylene absorbers, antioxidants, and antimicrobials from natural or artificial sources have been used to prevent food spoilage caused by moisture, air, bacterial growth, or adverse biochemical alterations due to oxidation [7,8]. To tackle the serious health issues associated with synthetic preservatives, the use of natural and functional food additives in active packaging to prolong the storage life of fresh food products attracted increasing attention in recent years.

The use of edible films and coatings in food packaging is another approach that has been adopted for a long time. The historic origins of this type of packaging can be traced back to 12th century China, where wax was used to retain moisture and provide a shiny appearance for citrus fruits. Later, lard or fat was used in England to prolong the keeping quality of meat-based products [9,10]. Noteworthy innovations in edible packaging research emphasized the development of environmentally friendly and biodegradable alternatives to the traditional petroleum-based polymers used in food packaging and preservation. As thin protective layers, edible films and coatings have been extensively investigated for food packaging and can be safely consumed despite being an integral component [11]. Films are applied as individual materials to cover and package food products as opposed to coatings, which are either directly applied to or formed on foods. By combining aspects related to food, preservative additives, and packaging, advancements in this field have steered the art of packaging in a sustainable direction. Films and coatings produced from various food-grade polymers, such as polysaccharides, lipids, and proteins, are being explored as environmentally friendly replacements for conventional food packaging. The effective implementation of novel materials and advanced methodologies can ensure the preservation and quality of food products by preventing moisture loss, lipid oxidation, off-odors, and color loss as well as enhance shelf life [12].

In the digital era, consumer awareness and preferences for safe and nutritious food consumption have significantly changed due to the increasing recognition of precautionary measures and health issues. One innovative approach for mitigating the food deterioration caused by undesirable means or the foodborne outbreaks driven by microbial pathogens involves incorporating active constituents into the packaging matrix and their controlled release to the contained food [3]. These active components can be antioxidants, antimicrobials, flavorings, colorants, nutrients, extracts, or other functional additives and can be either synthetically or naturally derived. However, emphasis has been placed on compounds of natural origin because certain synthetic compounds have been linked to adverse health effects. Commonly used natural compounds include those derived from plants, animals, microflora, spices, herbs, and essential oils. The European Commission (EC) Regulation No. 1935/2004—a regulatory framework that was established in the European Union in 2004—applies to all food contact materials (FCMs), including active packaging and edible films/coatings. Regulation No. 1935/2004 and all of its amendments establish comprehensive standards for FCMs and are considered their primary regulatory framework, whereas Regulation No. 2023/2006 formulates good manufacturing practices for ingredients and substances that are intended to interact with food products [9,13]. Because European Union regulations define a food additive as “any material that is not normally used as a characteristic constituent of food and not commonly consumed as a food in itself, whether or not it has nutritional content”, it is worth emphasizing that food additives are compounds that are introduced to foods during their manufacturing, processing, preparation, treatment, packaging, transport, or storage for technological purposes and eventually become a part of the food. Therefore, even if they are indirectly applied to food, the compounds present in active packaging and edible films/coatings must conform to food additive regulations. Regulation No. 450/2009, which comprehensively addresses the elements delivering active or intelligent features to packaging materials, also widely regulates active packaging [14,15].

Recent consumer demands and needs stimulated the development of active packaging and edible films/coatings with biopolymers and naturally occurring bioactive substances. This is in response to increasing health-related and environmental concerns regarding the use of chemical additives, as well as the accumulation of waste derived from synthetic packaging materials. However, natural materials are generally regarded as appealing value-added ingredients due to their diverse range of bioactivities. Among these, phenolic compounds play an important role because they are well known for enhancing the functional aspects of packaging materials via their antioxidant and antibacterial activities. Flavonoids, tannins, phenolic acids, lignans, and stilbenes are a few principal bioactive phenolic substances used in food packaging. The use of natural bioactive phenolic compounds with antimicrobial and antioxidant characteristics (Figure 1) in the production of active-packaging films and edible coatings significantly increased in recent years. Active packaging and edible films/coatings are part of innovative approaches for preserving fruits, vegetables, dairy products, fish, and meat products, as well as other processed food products. These packaging strategies can increase the shelf life of food products by inhibiting moisture loss; delaying bacterial growth; reducing fat, protein, and color oxidation; and improving shelf life.

In this review, significant improvements in the active packaging and edible films/coatings incorporated with phenolic compounds due to their antioxidant and antibacterial characteristics are discussed. Furthermore, phenolic compounds are briefly introduced with a focus on their beneficial antioxidant and antibacterial properties. Subsequently, state-of-the-art phenolic compounds and their applications in different active-packaging systems and edible films/coatings are surveyed. Finally, the safety concerns and future trends of phenolic-compound-based active packaging and edible films/coatings are highlighted by considering their novelty in the packaging domain.

## 2. Phenolic Compounds

Phenolic compounds, also referred to as polyphenols, are a large class of phytochemicals that are produced by plants as secondary metabolites. These compounds hold a distinct place among natural products because of their occurrence in a wide variety of foods, including fruits, vegetables, herbs, tea, coffee, chocolate, and wine. Chemically, phenolic compounds are a group of aromatic organic compounds that have at least one hydroxyl group directly connected to a benzene ring [16]. Polyphenols are typically categorized by their origin, structural diversity, and biological function into bioactive components such as flavonoids, tannins, phenolic acids, lignans, stilbenes, lignins, and coumarins. Over 10,000 phenolic compounds with a wide range of functionalities and structural diversity have been identified to date, with flavonoids being the largest group [17]. Table 1 summarizes the basic skeleton, main sources, and characteristics of the phenolic compounds.

The increasing consumer awareness of health-related and environmental benefits of using naturally derived materials and additives in packaging sparked extensive research and innovation in the field of food packaging. To realize adequate food packaging, food products must be effectively protected against moisture, oxygen, biochemical changes, and microbial deterioration. Among these destructive factors, oxidation in the packaging and microbial growth significantly deteriorate the shelf life of food products. To address the environmental concerns of conventional packaging waste used for preserving food products, techniques such as the use of naturally derived additives with antioxidant and antimicrobial properties emerged for developing active packaging and edible films/coatings. The functional properties of phenolic compounds in terms of their antioxidant activity and antibacterial function have been well established [5,10]. Phenolic compounds, which often exhibit antioxidant and antimicrobial characteristics, show promise as ingredients in active packaging and edible films/coatings due to their unique molecular structure. They maintain the physicochemical properties of food products, improve their sensory attributes, and protect them from oxidation. Furthermore, they can prolong the shelf life of food by preventing microbial development due to their antibacterial characteristics. The two primary characteristics of phenolic compounds—antioxidant and antimicrobial activities—are further discussed in the following subsections to help comprehend the significance of phenolic compounds in active packaging and edible films/coatings.

### 2.1. Antioxidant Activity

Food can undergo oxidative changes that lead to lipid rancidity, off-flavors, and a loss of color and flavor. Consequently, the nutritional quality and safety of food are compromised by the development of secondary, potentially toxic compounds [24]. Therefore, antioxidants must be added to maintain flavor and color while preventing metabolic alterations. In general, antioxidants inhibit or delay the oxidation in food by limiting the initiation or propagation of oxidative chain reactions. EC Regulation No. 1333/2008, which governs the use of food additives, defines antioxidants as “substances which extend the shelf-life of foods by preserving them against deterioration induced by oxidation, such as lipid rancidity and color changes” [25]. Among synthetic food additives, butylated hydroxy anisole, butylated hydroxytoluene, propyl gallate, and tert-butyl hydroquinone are commonly used as antioxidants to preserve food products. However, the use of natural antioxidants, including tocopherol, plant extracts, and essential oils from herbs and spices is an alternate strategy that is currently the subject of extensive research [19]. In particular, tocopherol, generally known as vitamin E, is an excellent radical-chain breaker in unsaturated fatty foods and is a lipid-soluble antioxidant that may be derived from food sources such as palm oil, sunflower, and soybeans. Commercially, tocopherols are the most extensively used antioxidants for preventing lipid oxidation in food products and exhibit a tremendous degree of complexity in terms of the various chemical and physicochemical factors involved. Furthermore, tocopherols are employed as antioxidants in food products in four different forms (α, β, γ, and δ), and their effectiveness decreases as follows: δ > γ > β > α [26,27].

Among the major naturally derived, functional compounds identified for use in active packaging and edible films/coatings, phenolic compounds (flavonoids, tannins, stilbenes, and phenolic acids), in particular, exhibit excellent antioxidant properties [9]. These compounds have become increasingly popular in recent years due to their antioxidant activities, especially because of their sources and their interactions with biopolymers. Polyphenols have been found to function via mechanisms such as free radical scavenging, single-electron transfer, hydrogen atom transfer, and metal chelation [3,9]. To extend the shelf life and improve the quality of food, active compounds are introduced to films/coatings or created as individual contraptions (sachets, pads, or labels) that can either release or absorb reactive radicals [28]. Numerous phenolic compounds found in fruits, vegetables, herbs, tea, coffee, chocolate, and wine have been reported as strong antioxidants. For example, fruits such as kiwi, prunes, olives, berries, cherries, and citrus fruits have been proven to have antioxidants with significant activity [29,30,31,32,33,34,35,36,37,38,39,40].

### 2.2. Antimicrobial Activity

Consumers appear to be significantly concerned about foodborne illnesses, particularly in the current era with considerably higher packaged food consumption. Therefore, antimicrobial compounds are individually or collectively applied to food or packaging materials [41]. According to Regulation No. 1333/2008, antimicrobial compounds are described as “compounds which extend the shelf-life of foods by protecting food products against deterioration caused by microorganisms and/or which protect the food products from the growth of pathogenic microorganisms” [9]. The use of antimicrobial packaging systems is an excellent approach to inhibit the activity of specific microorganisms and prevent the growth of foodborne pathogens through the formation of a comprehensive and effective barrier. Several methods exist for incorporating antimicrobial compounds into antimicrobial packaging systems, including fabricating films with antimicrobial substances, directly integrating antimicrobial compounds into packaging films, or coating packaged films [42,43,44]. The application of an antimicrobial compound (migrating or non-migrating) and the potency of its interactions with the packaging and food matrix determine the efficacy of an antimicrobial packaging system. Two theories explain the effectiveness of these systems: (1) the antimicrobial compound reaches the surface of the food (migrating film) and (2) the compound significantly inhibits microbial surface growth without migrating (non-migrating film). Furthermore, the direct addition of synthetic antimicrobial compounds to foods can effectively limit the spread and viability of several microorganisms. Nevertheless, consumers prefer naturally processed and preservative-free food products with an extended shelf life.

The development of packaging materials with natural antimicrobial agents has become increasingly popular. Natural ingredients, such as bacteriocins, enzymes, and plant-derived compounds, are biologically derived components that have been employed in antimicrobial packaging. Plant-derived compounds are mainly secondary metabolites that exhibit several advantages such as antimicrobial properties against harmful and spoilage microbes. Polyphenols, phenolic acids, flavonoids, tannins, quinones, coumarins, terpenoids, and alkaloids are the major classes of compounds responsible for the antimicrobial action. Numerous naturally occurring phenolic compounds that are present in various plant sources such as fruits (apple, grape, pomegranate, and orange); vegetables (cabbage and onion); herbs (garlic, oregano, thyme, and rosemary); and spices (pepper, cardamom, and clove) have been documented to have antimicrobial properties [45,46,47,48,49,50]. Although the efficacy of natural antimicrobials has been demonstrated in laboratory settings, challenges remain in ensuring their effectiveness in practical applications for foods under different environmental conditions [51].

## 3. Phenolic Compounds in Food Packaging

### 3.1. Flavonoids and Tannins

The majority of phenolic compounds are flavonoids, which include over 8000 different compounds grouped into subclasses of flavanones, flavonols, flavanones, isoflavones, flavanols, quercetin, and anthocyanins [19,52]. The basic structural component of flavonoids typically includes three hydroxyl groups and hydroxylated phenolic compounds with a C6–C3–C6 link in the aromatic ring [20]. Flavonoids with antimicrobial, antioxidant, anti-infective, and antifungal activities are abundant in nature and are derived from a wide range of sources such as berries, herbs, cacao, grapes, green and black tea, citrus fruits, spinach, soybeans, olives, cherries, and red wine. Incorporating flavonoids into packaging materials is an effective strategy to enhance the safety of packaged foods and preserve their quality. These compounds can inhibit microbial growth by releasing antioxidizing agents [53]. Flavonoids are increasingly preferred for use in active packaging and edible films/coatings; however, their stability (due to the extraction process and storage conditions) makes their use as active ingredients challenging.

Tannins are water-soluble, astringent, and complex phenolics that are available from diverse sources such as tea, coffee, chocolate, berries, apples, and wine. They can be classified into hydrolyzable and condensed tannins based on their resistance to hydrolysis [21]. Because tannins are regarded as generally recognized as safe (GRAS) additives by the US Food and Drug Administration (FDA), the characteristic features of tannins, including antimicrobial, antifungal, antioxidant, and UV absorption properties, can be readily exploited for active-packaging applications [3]. In a study on the development of active-packaging films, the incorporation of tannin was found to be effective in improving the UV-blocking and antioxidant properties of tannin-cellulose films [54]. Similarly, tannin-containing films composed of chitosan and cellulose demonstrated significant antioxidant and antibacterial activities [55].

### 3.2. Phenolic Acids—Hydroxybenzoic and Hydroxycinnamic Acids

Phenolic acids are non-flavonoid polyphenolic substances that are present in various food sources and are characterized by a carboxyl group connected to a benzene ring [18]. Natural phenolic acids are categorized into two main groups based on the number of carboxylic acids, hydroxybenzoic (C6–C1) and hydroxycinnamic acids (C6–C3), which are derived from benzoic and cinnamic acids, respectively [21]. They are the simplest class of phenolics and serve as building blocks for several other compounds. Gallic, sinapic, and ellagic acids are some of the main hydroxybenzoic acids, whereas caffeic, p-coumaric, and ferulic acids are the main hydroxycinnamic acids. Numerous studies have found that phenolic acids exhibit biological activities, such as antioxidant, anti-inflammatory, antibacterial, and other functional characteristics. Moreover, phenolic acids are considered desirable food preservatives because they significantly inhibit the development of numerous harmful bacteria and fungi, including *E. coli*, *Bacillus cereus*, *Staphylococcus aureus*, *Aspergillus flavus*, and *Aspergillus parasiticus* [21,56,57]. For instance, gallic acid, which exists in high amounts in red wine, tea leaves, berries, mango, citrus fruits, and soy, is recognized primarily for its antioxidant properties as well as antibacterial and anti-inflammatory properties [58]. Similarly, ellagic acid, which exhibits an array of biological activities, is a significant polyphenol antioxidant found in several fruits, nuts, and seeds. It contains potential biomolecules with interesting biological properties such as antioxidant, antimicrobial, and UV-barrier characteristics [59]. Notably, numerous phenolic acids have been employed as valuable components in active packaging and edible films/coatings to achieve sustained antimicrobial, antioxidant, and other functional characteristics.

The effective incorporation of gallic acid has been achieved by electrospinning it into hydroxypropyl methylcellulose nanofibers, which were found to be an effective active-packaging material for delaying oxidation during the storage of walnut [56]. Similarly, the antioxidant capacity of gallic acid to scavenge free radicals has been shown to delay lipid oxidation in numerous edible coating materials [60,61,62]. Phenolic acid, ellagic acid, and chitosan have been used to fabricate active food-packaging films with high antioxidant, antimicrobial, and UV-light-resistance characteristics [63]. Furthermore, numerous studies have shown that the addition of phenolic acids, such as caffeic acid and p-coumaric acid, to fatty food products, such as processed fish and fish products, delays or prevents oxidative degradation [64,65,66]. Hydrocolloid films prepared from chitosan and fish gelatin and filled with the naturally occurring antioxidants of caffeic and p-coumaric acids demonstrated excellent results in terms of preventing the oxidation of fatty food products [67]. Additionally, films containing caffeic acid had higher levels of chelating iron, reducing power, and antioxidant activity than those of films containing p-coumaric acid.

### 3.3. Lignans

Lignans are secondary plant metabolites with complex phytoestrogen-related chemical compositions. Their basic structure includes a combination of phenylpropanoid dimers (C6–C3) connected by the central carbons of the side chains [19,20]. Lignans are primarily derived from oilseeds of flaxseed and sesame, legumes, whole grains, and several types of berries. Similarly to several other phenolic compounds, lignans exhibit a wide range of bioactivities and have been used by humans for a long time. Due to their significant bioactivity as antioxidants and antimicrobials, lignans have been used in food science and nutrition since ancient times. Lignans have antioxidant capacities because they function as hydrogen donors and complex divalent transition metal cations [68].

### 3.4. Stillbenes

Stilbenes are a class of plant polyphenols that received significant interest because of their complex chemical structures and wide range of biological functions. The basic structure of stilbenes is C6–C2–C6, which has two benzene rings joined by a double bond [19,69]. Food sources that contain stilbenes include grapes, pine, almonds, peanuts, sorghum, berries, and wine. The use of stilbenes is attracting attention because of their potential bioactive components. For instance, the stilbene resveratrol has been shown to possess a range of bioactive properties, such as antioxidant, anti-inflammatory, and antibacterial activities [70,71].

Cellulose bilayer films incorporated with resveratrol have been prepared. These films showed antimicrobial activity against Campylobacter and exhibited potential bioactive packaging properties for improving food safety [71]. Similarly, a study on assessing the antioxidant and antibacterial properties of carboxymethyl cellulose films containing resveratrol and eugenol suggested that the addition of resveratrol and eugenol to the films increased their total phenolic content, free-radical-scavenging activity, reducing power, and antibacterial activities [72].

### 3.5. Lignins

The phenolic chemicals present in lignins, which are naturally occurring antioxidants, are derived from numerous renewable sources such as spruce, jute, cotton, hemp, pine, birch, and agricultural crops [73]. Due to their functional groups, lignins have an aromatic, highly cross-linked structure and are particularly reactive. Consequently, they can combine with various polymers to modify their morphological, hydrophilic, and strength-related properties. The considerable antioxidant activity is caused by the properties of lignins induced by their phenolic hydroxyl groups, aliphatic hydroxyl groups, low molecular weight, and narrow polydispersity [74,75]. Because lignins are typically immiscible, their incorporation into polymer matrices requires considerable time and has limited practical applications and industrial-scale use [75]. However, the influence of lignins on the physicochemical and functional characteristics of natural polymers in the preparation of various films has been thoroughly investigated in the recent past.

Lignin nanoparticles have been used as a filler to investigate the antioxidant and antibacterial characteristics of films developed using polyvinyl alcohol/chitosan for active-packaging applications [76]. The antioxidant capacity and antimicrobial performance of the films enhanced over time, inhibiting the growth of Gram-negative bacteria, including *Xanthomonas arboricola* pv. *pruni* and *Erwinia carotovora* subsp. *carotovora*. Moreover, lignins have been found to be applicable as a competitive material for active food packaging. A poly(lactic acid) composite films with lignin showed excellent UV resistance and enhanced antioxidant capacities [77].

### 3.6. Coumarins

Coumarin is a naturally occurring phenolic compound found in several plants, such as cinnamon, cloves, tonka beans, celery, and apricots. It consists of an aromatic ring fused to a condensed lactone ring and has a spicy, fresh-hay, or vanilla fragrance [78]. Coumarin exhibits bioactive characteristics such as anticoagulant, antimicrobial, antifungal, and antioxidant effects. For instance, chitosan and fish gelatin have been used to create coumarin-containing bioactive films, which have demonstrated antioxidant properties. Films containing coumarin were found to release more free radicals than a control film [79]. Contrary to other phenolic compounds, the use of coumarin in active packaging and edible films/coatings has not been investigated, providing a good opportunity to investigate and demonstrate its active qualities.

## 4. Application of Phenolic Compounds in Packaging Films/Coatings

Natural phenolic compounds are ideal candidates for food packaging because they exhibit antioxidant and antibacterial activities along with a variety of relevant properties such as UV-light resistance, color, flavor, and sensory attributes [80]. They are selectively applied as food additives and preservatives in active packaging and edible films/coatings to extend the shelf life of packaging products by ensuring appropriate oxidative and microbial environments. The packaging matrix can release specific phenolics into the ambient environment gradually or at a controlled rate (in the packaging headspace or on the surface of food products), inhibiting the further oxidation or microorganism growth [81]. Considering the increasing interest in the incorporation of phenolic compounds into active packaging and edible films/coatings for food products, the following sections comprehensively discuss their applications.

### 4.1. Active Packaging

In response to dynamic changes in market trends and consumer preferences for convenient, safe, healthy, and high-quality food products, innovations in food packaging systems continually evolved with the development of modern packaging techniques. For instance, active packaging—a cutting-edge approach to food packaging—evolved in response to the rising consumer demand for fresh food products with extended shelf life. In this approach, the environment, product, and packaging interact simultaneously to enhance shelf life without compromising safety, quality, and sensory attributes. To improve or protect the integrity of food products or to prolong their shelf life, active packaging incorporates components that can release or absorb compounds into or from the packaged products or their surroundings [7,82]. Examples of active food-packaging agents include oxygen scavengers, antioxidants, and antimicrobial agents. These materials enable the release or absorption of substances from packaged food or its surroundings [83].

Active agents are primarily incorporated into or applied onto the surfaces of packaging materials as sachets, labels, or pads. They may be natural or synthetic in origin and used to achieve a range of specific objectives, such as antibacterial or antioxidant effects. However, the use of natural antioxidants or antimicrobial agents in food products is attracting greater attention due to the potential adverse effects associated with the intake of synthetic-type active additives and preservatives [84,85]. Fruits, vegetables, herbs, spices, seeds, tea, chocolate, and wine are major food sources that naturally contain antioxidants and antibacterial agents in the form of phenolic compounds. Furthermore, because the majority of them may be derived from underutilized plant species and food byproducts, the spotlight on these natural ingredients is motivated not only by their bioactive constituents but also by their potential economic value [86]. Figure 2 illustrates the concepts related to active packaging, with a focus on the antioxidant and antibacterial properties of phenolic compounds.

Active packaging incorporated with phenolic compounds and their impact on packaging operation have been comprehensively analyzed in the recent past. In particular, numerous studies have been conducted on extending the shelf life and improving the quality of food products. These investigations have been conducted worldwide using different polymers and phenolic compounds for various purposes. Table 2 lists several naturally occurring phenolic compounds that have been directly incorporated into polymers for active-packaging applications. For example, chitosan and kombucha-tea-based active films have shown improved antioxidant activity and antimicrobial properties [87]. By delaying lipid oxidation and microbiological growth, the developed active-packaging film with various phenolic components, primarily catechin, effectively functioned as active packaging and extended the shelf life of minced beef. Gallic acid, a phenolic compound derived from apple pomace, has been incorporated into a polyvinyl alcohol matrix to develop active antioxidant-integrated food-packaging films, which demonstrated their ability to delay lipid oxidation in soybean oil [88].

Tannins, which are naturally occurring phenolic compounds that can be derived from a range of fruits and vegetables, including tea, coffee, chocolate, and wine, are known to exhibit antioxidant characteristics due to their phenolic hydroxyl groups [54,97]. For instance, sachets created using chitosan-based active film materials and tannin extracted from chestnut have been found to be beneficial for preserving fresh pasta for 60 days [89]. Tannins have also been employed as antioxidants in the development of active-packaging materials [98,99,100]. Thymol-enriched jackfruit peel waste has been used to fabricate active films from tapioca starch, which exhibited antimicrobial properties and the ability to preserve cherry tomatoes [90]. Similarly, ground meat and sunflower oil have been effectively preserved by antioxidant and antibacterial active-packaging films made from corn starch containing anthocyanin from red cabbage extract as well as corn starch containing eugenol [91,92]. Therefore, phenolic compounds are effective natural additives that delay oxidative processes and the growth of microorganisms. Consequently, they facilitate the development of fresh, healthy, and safe food products with a prolonged shelf life.

### 4.2. Edible Films and Coatings

Due to issues such as littering, waste incineration, CO_2_ production, and the release of toxicants into the environment, conventional packaging materials are typically single-use or difficult-to-recycle materials that pose significant socioeconomic and environmental concerns [101,102]. Establishing more sustainable, high-quality, and healthier food-packaging systems is a primary concern for both the food industry and consumers. Consequently, edible films/coatings are being extensively explored for food preservation and packaging as well as to address sustainability concerns and consumer expectations for natural, healthy, and safe foods [13]. In general, edible packaging refers to readily consumable packaged-food products with films or a thin layer (coating) on the surface of the food matrix. In this context, films are individual layers that can be used to wrap food products, whereas coatings are directly applied to or formed on the surfaces of foods. More specifically, they may be formed between different food product components or on the surface of the product. Because food products comprising edible films and coatings are intended to be consumed directly, they are mainly produced using edible biopolymers and food-grade additives [103]. Additionally, they must adhere to good manufacturing practices while being processed and applied to food products. In particular, the biopolymers and additives used to prepare edible films and coatings must be conferred with the GRAS status established by the US FDA [10].

Natural additives are attracting greater attention than their synthetic counterparts, as synthetic additives can potentially take a long time to fully degrade, release contaminants and toxins, and possibly pose a risk to food safety and health. The use of phenolic compounds as bioactive components in active edible packaging has been found to be effective, because they are believed to exhibit antioxidant and antibacterial activities [10,104]. The incorporation of these compounds stimulates physical, mechanical, and biological activities, which improve the quality, sensory appeal, and shelf life of food products. The mechanisms through which the functional properties of edible films/coatings are enriched by the inclusion of phenolic compounds are illustrated in Figure 3. Certain naturally occurring phenolic compounds, such as those found in fruits, vegetables, spices, tea, herbs, and wine, can be used as feasible and efficient sources of strong active compounds, such as flavonoids, tannins, polyphenols, anthocyanins, stilbenes, and pigments, in edible films/coatings and can replace synthetic additives and preservatives.

Improvements in edible films/coatings incorporated with phenolic compounds and beneficial effects on the packing performance have been achieved over time, similarly to those of active packaging. Studies conducted on incorporating phenolic compounds into edible films/coatings to bestow them with antioxidant and antibacterial characteristics are listed in Table 2. These compounds are effective in preserving the quality and shelf life of various food products. For instance, coatings comprising chitosan and green tea extract containing phenolic compounds enriched with catechin, and epicatechin gallate prolongs the preservation of walnuts by delaying lipid oxidation and fungal growth [91]. Principally, the phenolic components of green tea extracts function as antioxidants to hinder the initiation of a radical cycle, bind to catalysts composed of transition metal ions, and interact with free radicals to prevent lipid oxidation [105,106]. A composite coating fabricated using whey protein concentrates, glycerol, and carboxymethyl cellulose with the gallic acid derived from rosemary extract significantly enhanced the color and oxidative stability of stored sunflower seed kernels [94]. Furthermore, the fabrication of a chitosan–pullulan edible composite coating that included pomegranate peel extracts preserved the sensory qualities and extended the shelf life of bell peppers because of gallic acid, ellagic acid, and other phenolic components [95]. Two phenolic-type antioxidants—hydroxytyrosol and 3,4-dihydroxyphenylglycol—which are naturally found in olives, were added to beef meat wrapped in pectin/fish-gelatin edible films and observed to improve quality and shelf life [96].

Although edible films/coatings offer a range of benefits, certain obstacles prevent their commercial use. For instance, edible films/coatings must be capable of serving their function safely and effectively during the life of the product and remain stable under different atmospheric conditions [10]. Other issues related to the efficacy of edible films/coatings include their poor moisture-barrier properties due to the hydrophobic nature of the majority of edible packaging materials, poor temperature regulation, and relative humidity control. Because of the lack of an approved standard dosage for the application of various edible films/coatings, regulation and safety-related issues additionally hinder their use [107]. However, the use of phenolic compounds as carriers of active ingredients in the fabrication of edible films/coatings is a promising strategy. Phenolic compounds, which are abundantly present in several food sources and play a key role in the development of sustainable chemical-additive-free packaging materials, are becoming increasingly important as consumers seek a greater variety of fresh, natural products with extended shelf life.

## 5. Conclusions and Future Perspectives

Novel food-packaging techniques, such as active packaging and the use of edible films/coatings, show tremendous promise in terms of enhancing food quality and safety, thereby contributing to the trend toward environmentally sustainable and smart packaging approaches. The abundant, naturally available phenolic compounds derived from numerous sources demonstrate remarkable promise as alternative food additives/preservatives to synthetic chemical additives for use in active packaging and edible films/coatings due to their antioxidant and antimicrobial characteristics. With increasing consumer preferences towards naturally occurring and minimally processed preserved foods, phenolic compounds have been gaining substantial limelight for food packaging applications. Natural phenolic compounds have also shown other functional improvements to date in terms of their applications in food packaging. Recent developments in active packaging and edible films/coatings with a focus on human health and environmental aspects further revolutionized the inclusion of phenolic compounds in packaging domain. In this context, active packaging necessitates a reconsideration of the fundamental notion regarding the lack of interactions between packaging materials and food products. Additionally, ensuring the use of safe food-grade additives to fabricate edible films/coatings and applying them to food products are crucial.

Although novel uses of phenolic compounds as natural additives have great potentials to improve food safety and quality, more research is still required to address significant obstacles such as stability, interaction, and retaining other desired packaging attributes. While using phenolic compounds, there is a need of adequate studies to estimate their affinity and stability while exhibiting functional characteristics within packaged product. Moreover, the challenges are to maintain good barrier and mechanical properties of the packaging materials. Other limitations of phenolic compounds in packaging applications are their limited resistance to high temperatures and long-term use. Therefore, they must be employed accordingly to avoid degradations and loss of properties. Additionally, when used in application, it is important to take into account the intense organoleptic properties of specific phenolic compounds that may affect the sensory qualities of food.

In conclusion, this review provides an overview of the current state of the use of phenolic compounds in active packaging and edible films/coatings in the pursuit of sustainable and smart packaging, with a special emphasis on investigating alternatives to synthetic additives and implementing natural and safe additives in food packaging applications. In the near future, environmentally friendly packaging materials incorporated with natural ingredients are anticipated to enable the preparation of minimally processed packaged food products. Phenolic-compound-incorporated active packaging and edible films/coatings have the potential to prevent lipid oxidation, discoloration, and microbiological degradation in food packaging. Therefore, phenolic compounds offer a promising opportunity in the field of innovative packaging as a means of satisfying consumer demand for nutritious and fresh food products while reducing reliance on synthetic additives.

## Figures and Tables

**Figure 1 molecules-27-07513-f001:**
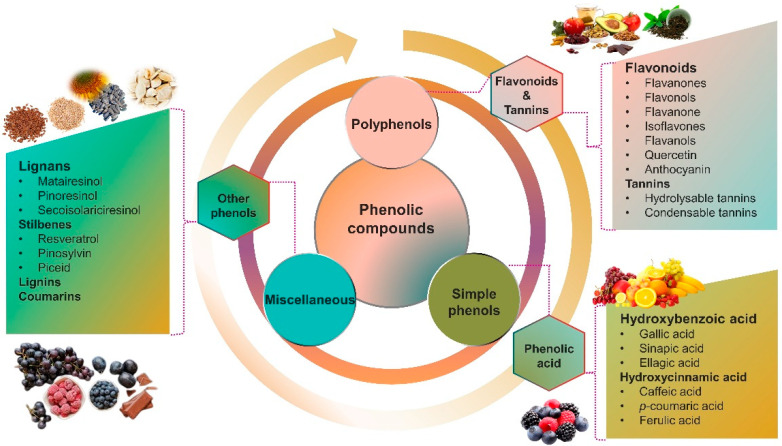
Schematic illustrating numerous naturally occurring phenolic compounds.

**Figure 2 molecules-27-07513-f002:**
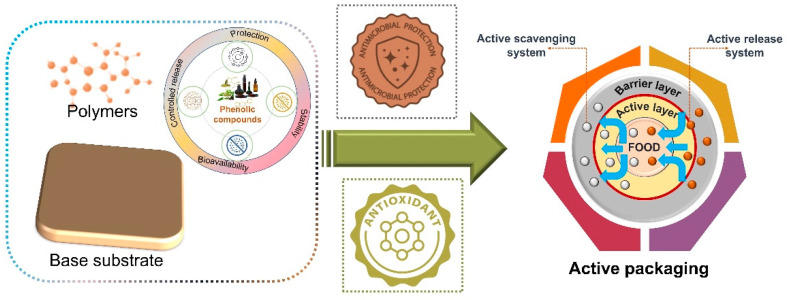
Graphical depiction of antioxidant and antimicrobial properties of phenolic compounds in the development of an active-packaging system.

**Figure 3 molecules-27-07513-f003:**
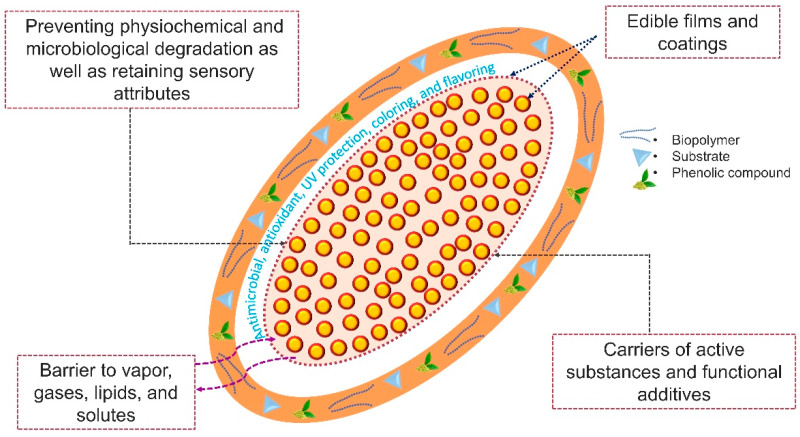
Enhancements in the characteristics of edible films/coatings containing phenolic compounds.

**Table 1 molecules-27-07513-t001:** Classification, basic skeleton, sources, and characteristics of phenolic compounds.

Phenolic Compounds	Basic Skeleton	Main Sources	Characteristics	References
Flavonoids (flavanones, flavonols, flavanone, isoflavones, flavanols, quercetin, anthocyanin)	C6–C3–C6	Wide range of sources (berries, herbs, cacao, grapes, green and black tea, citrus fruits, spinach, soybeans, olives, cherries, and red wine)	Antioxidant, antimicrobial, anti-infective, and antifungal activities	[3,18]
Tannins (hydrolysable tannins, condensable tannins)	(C6–C1)_n_	Tea, coffee, chocolate, berries, apples, and wine	Antimicrobial, antifungal, and antioxidant capabilities in addition to UV absorption	[19]
Phenolic acids (hydroxybenzoic acid (gallic acid, sinapic acid, ellagic acid) and hydroxycinnamic acid (caffeic acid, p-coumaric acid, ferulic acid)	C6–C1 and C6–C3	Berries, persimmon, apple juice, grapes, mustard, oranges, rye, coffee, mushrooms, propolis, tea, and wine	Antioxidant, antimicrobial, and anti-infection activities	[20]
Lignans (matairesinol, pinoresinol, secoisolariciresinol)	C6–C3	Oilseeds such as flaxseed, sesame, legumes, whole grains, and berries	Antioxidants and antimicrobial properties	[21]
Stillbenes (resveratrol, pinosylvin, piceid)	C6–C2–C6	Grapes, pine, almond, peanuts, sorghum, berries, and wine	Antioxidant, and anti-infective, activities	[18,20]
Lignins	(C3–C6)_n_	Spruce, jute, cotton, hemp, pine, and birch	Antioxidants and antimicrobial properties	[22,23]
Coumarins	C6–C3	Cinnamon, cloves, tonka bean, celery, and apricots	Antioxidants and antimicrobial properties	[20]

**Table 2 molecules-27-07513-t002:** Characteristics, and application of active-packaging materials using phenolic compounds from different sources.

Packaging Type	Polymers	Phenolic Compounds	Structure	Sources	Characteristics	Applications	Reference
Active film	Chitosan	Catechin	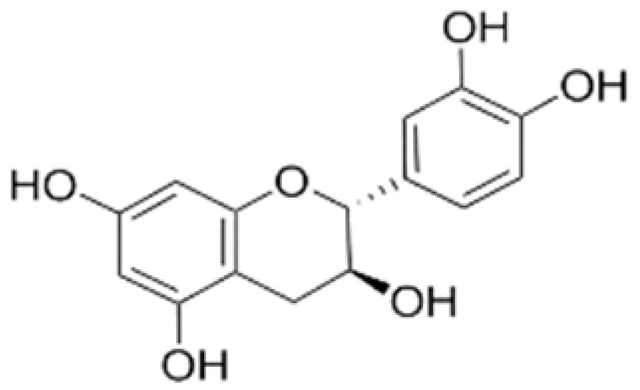	Kombucha tea	Retardation of lipid oxidation inhibited microbial growth of *E. coli* and *S. aureus*, enhanced the antioxidant activity, and increased the protective effect of the film against ultraviolet.	Minced beef	[87]
Active film	Polyvinyl alcohol	Gallic acid	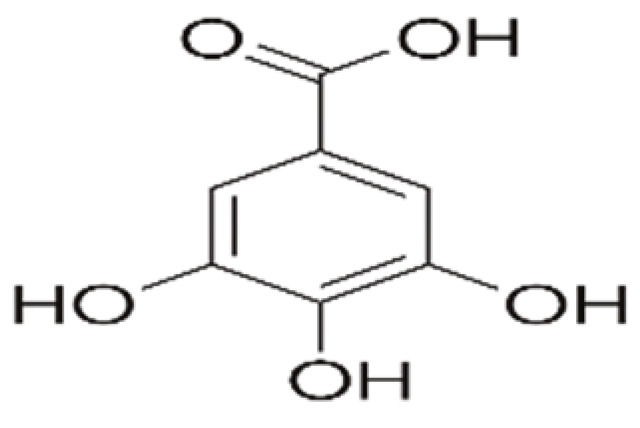	Apple pomace	Exhibited antioxidant capacity in delaying lipid oxidation.	Soybean oil	[88]
Active film	Chitosan	Tannin	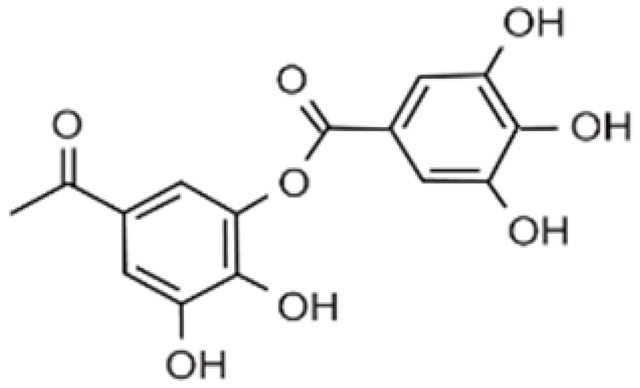	Chestnut extract	Prevented microbial growth on the surface of pasta for 60-day storage.	Fresh pasta	[89]
Active film	Tapioca starch	Thymol	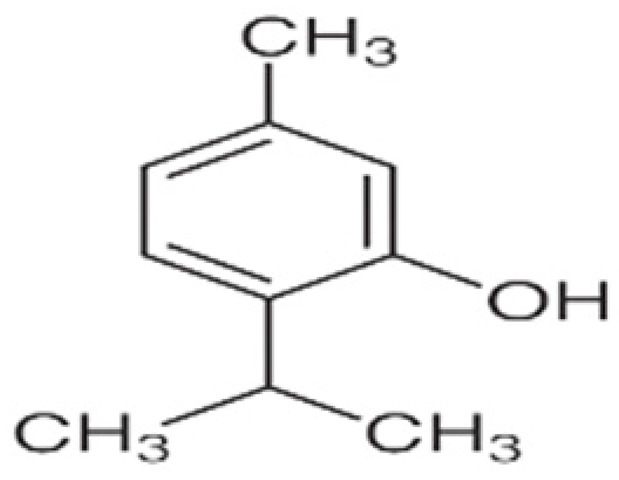	Jackfruit peel	Inhibitory effect against *E. coli* and *S. aureus*, improvement in water vapor permeability, flexibility, and reduction in water solubility.	Cherry tomato	[90]
Active film	Corn starch	Eugenol	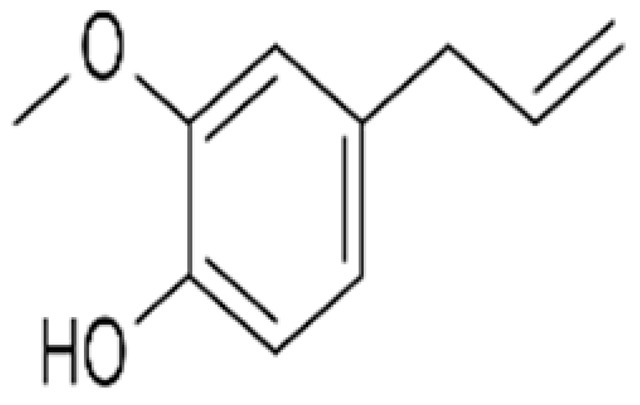	Clove and cinnamon oil	Effective in preventing sunflower oil oxidation in accelerated storage conditions.	Sunflower oil	[91]
Active film	Corn starch	Anthocyanin	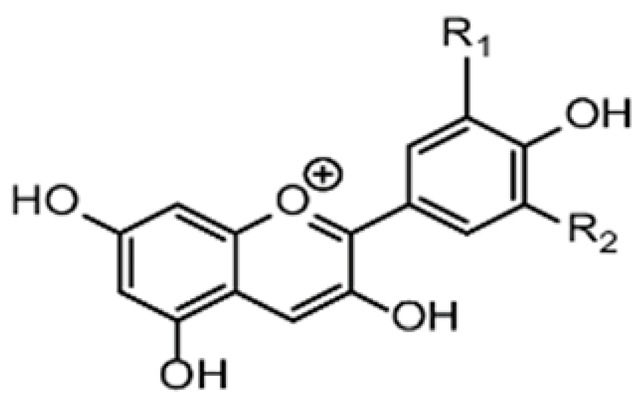	Red cabbage	Increased antioxidant activity with improved water vapor permeability and mechanical strength.	Ground beef	[92]
Edible coating	Chitosan	Catechin, Epicatechin gallate	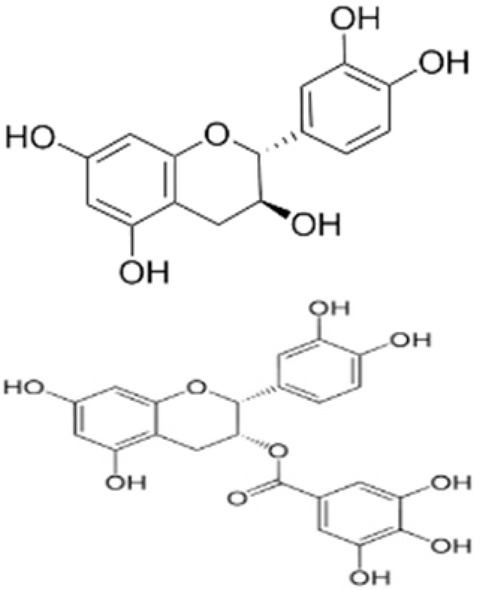	Green tea extract	Effective in lowering oxidation activity, fungal development, and preserving the sensory qualities.	Walnut kernels	[93]
Edible coating	Carboxymethyl cellulose and Whey protein concentrate	Gallic acid	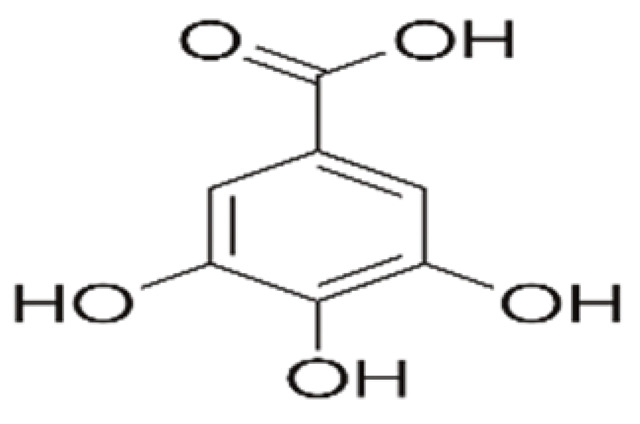	Rosemary extract	Desirable color and oxidative stability with formulated coating.	Sunflower seeds	[94]
Edible active film	Pectin-gelatin blending	Hydroxytyrosol (HT), 3,4-dihydroxyphenylglycol (DHPG)	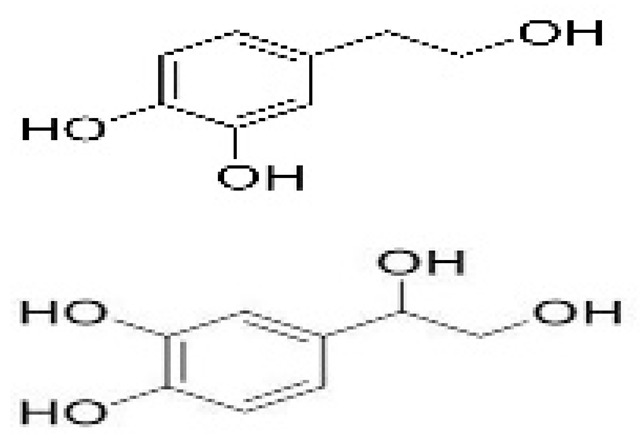	Olive oil extract	Exhibited antioxidant activity and delayed lipid oxidation of beef meat.	Beef meat	[95]
Edible coating	Chitosan/Pullulan	gallic acid, ellagic acid, and other phenolic compounds	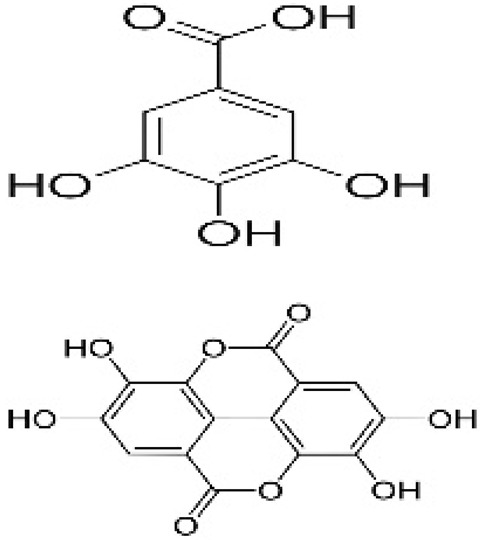	Pomegranate peel	Shelf-life extension and physiochemical quality maintenance over a storage period of 18 days.	Bell pepper	[96]

## Data Availability

Not applicable.

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
