# Peer review of "Phenolic Compounds in Active Packaging and Edible Films/Coatings: Natural Bioactive Molecules and Novel Packaging Ingredients"

_molecules, 2022, doi:10.3390/molecules27217513_

Round 1

Reviewer 1 Report

The document is quite interesting, I think the compilation is very good and concise. The development of the text takes you by the hand for complete understanding.

Author Response

Dear Reviewer,

I have resubmitted the review article entitled “Phenolic Compounds in Active Packaging and Edible Films/Coatings: Natural Bioactive Molecules and Novel Packaging Ingredients” to Molecules Journal after revising the manuscript according to the reviewer’s comments. Those comments are all quite valuable and helpful in revising and improving our work, as well as providing significant direction for our research. Authors carefully considered the comments and made improvements of the manuscript's quality.

We wish you to consider this manuscript for publication and extend your cordial support in this resubmission paper.

Sincerely,

Youn Suk Lee, Ph.D

Reviewer 2 Report

This article presents on the antioxidant compounds that can be used as fruit coatings and may help to provide edible and organic fruit coatings at industrial scale. Before recommending this article for publication, there are some shortcomings for that should be resolve.

Abstract

First sentence of the abstract is very long that’s why the meaning if the sentence is not clear. The authors are directed to write in short and correct sentences.

Findings of the review are missing in the abstract.

Line 19 write film or coatings.

Introduction

First sentence of the introduction must be revise.

Introduction line 40 lack reference.

In introduction section discuss about phenolic compounds specifically groups of phenolic compounds.

Line 156 must be cited.

Line 179 must be cited.

Line 180 add nanoparticles prepared with organic polymers.

Line 194 must be cited.

Line 213 could be cited with the following studies.

 https://doi.org/10.1002/aoc.5190, https://doi.org/10.1016/j.bcab.2020.101729,

Species names like bacteria names and fungal strain names must be italicized.

As a while the review article is well presented but mostly citations are missing, species names are not italicized.

Discuss about novel synthetic packaging molecules its harmful effects and benefits as well. Also justify with examples how phenolic compounds are better as compared to other compounds like alkaloids etc.  

Author Response

(The authors gave the same response as above.)
